# Quality Assessment of Raw Honey Issued from Eastern Romania

**Aida Albu, Cristina-Gabriela Radu-Rusu \*, Ioan Mircea Pop \*, Gabriela Frunza and Gherasim Nacu**

Animal Science Faculty, "Ion Ionescu de la Brad" University of Agricultural Sciences and Veterinary Medicine, 8 Mihail Sadoveanu Alley, 700489 Iasi, Romania; albu.aida@uaiasi.ro (A.A.); frunza.gabriela@uaiasi.ro (G.F.); gnacu@uaiasi.ro (G.N.)
\* Correspondence: cradurusu@uaiasi.ro (C.-G.R.-R.); popim@uaiasi.ro (I.M.P.)

**Abstract:** Romania is known among the main European honey producers, due to the variety of landforms as well as the diversity of the flora. Thirty-four honey samples of the acacia, linden and multifloral types, produced in eastern Romania and collected during 2013–2018, were physico-chemically analyzed using methods provided by the national and EU standards. The results of water-insoluble solids, color and refractive index were found to be 0.023–0.131%, 0.3–76.4 mm Pfund and 1.485–1.499, respectively. The moisture content ranged between 15.20% and 20.77%, solid substances content ranged between 79.23% and 84.80% and total soluble substances content ranged between 77.83 °Brix and 83.26 °Brix. The obtained values of specific gravity were from 1.414 to 1.450 g/cm$^3$, pH ranged from 3.673 to 5.503 and free acidity ranged between 2.4 meq and 50 meq kg$^{-1}$. The ash content and the electrical conductivity varied between 0.030 and 0.543% and 130 and 679 μS cm$^{-1}$, respectively. Pearson's correlation analysis showed an intense association of the ash content with electrical conductivity (r = 0.81). Our findings reveal the qualitative level of Romanian honey and the variation in quality parameters due to factors such as geographical region, climatic conditions, botanical origin and handling or storage conditions.

**Keywords:** raw honey; conductivity; specific gravity; total soluble substances; refractive index; quality





## 1. Introduction

Honey is known and recognized as a wholesome food consumed due to its extraordinary composition, in terms of nutritional and therapeutic features [1,2]. Honey was considered to have healing properties by the ancient civilizations going back in time from the Chinese Empire to the Egyptian Empire. Nowadays, it is quite a trend to investigate alternative natural foods and molecules, such as bee products, which have been supposed to trigger active pharmacologic and metabolic pathways and to generate beneficial effects on consumers' health. [3].

Romania has an ancient tradition of beekeeping and now is one of the most important honey producers in Europe, due to the variety of landforms as well as the diverse and very rich flora. In the Romanian flora, there is a series of species of honey plants that stand out through a high honey production. This country has potential to offer sufficient and variate botanical resources to the indigenous bees (*Apis mellifera carpatica*) in order to obtain a wide panel of unifloral and multifloral honeys, such as the linden (*Tilia tomentosa*) type, the acacia sort (*Robinia pseudoacacia*), the sunflower variety (*Helianthus anuus*) and, of course, the multifloral type (usually from spring–summer meadows and grasslands). North East Romania, specifically the historical province of Moldova, is known as the largest linden massive [4–6]. From the mountains to the plains, the eastern region of Romania is rich in cultivated and spontaneous polliniferous and nectariferous plants.

This food contains many important constituents such as sugars, acids, proteins, vitamins, enzymes and micro- and macro-elements [7]. The low water inner content is among the most important traits with relevance in maintaining honey freshness, stopping the development of yeasts and avoiding fermentative processes. Yeast and mold proliferation in honey could be regulated by the free acidity and water content within [8]. Both electrical conductivity and crude ash analyses are frequently used in honey quality inspections. Those traits, especially electrical conductivity, are considered very good criteria for assessment of the botanical origin and purity of honey [9]. Legislation has established maximum allowed values for some parameters, such as a maximum 20% moisture content, maximum 50 meq kg$^{-1}$ free acidity, maximum acceptable content of water-insoluble solids no more than 0.1% and no more than 0.5% for pressed honey and a maximum permitted value of 0.8 μS cm$^{-1}$ conductivity for nectar honey, with the exception of chestnut honey which should be not less than 0.8 μS cm$^{-1}$ [10,11]. Such physical and chemical traits are keystones on the way to assess honey quality and to identify its authenticity and origin. All honey parameters are correlated twice. Many studies have shown that the variation in one parameter leads to the modification of the other parameters such as ash and electrical conductivity and refractive index with specific gravity and total soluble substances.

In this work, studies were carried out to show the high quality of Romanian honey, quality that remains high regardless of the geographical area and year of harvest.

## 2. Materials and Methods

Thirty-four raw honey samples produced by *Apis mellifera* of three of the most known honey types: acacia (A) (twelve samples), linden (L) (ten samples) and multifloral (M) (twelve samples), were collected during 2013–2018 directly from beekeepers. The honey produced in eastern Romania comes from different sites of Iasi, Vaslui and Suceava counties (Figure 1a,b). The collection of honey samples was carried on throughout different years, from 2013 to 2018, during the period of maximum flowering of melliferous plants (acacia and linden). Raw honey samples were kept in glass jars, preserved at room temperature (20 ± 3 °C), away from light. The analyses were performed within a maximum of 2 weeks from the arrival of the samples to avoid their degradation as well as the modification of their parameters, taking into account the fact that they were raw honey samples and not commercial honey. The analyses were performed in triplicate. The crystallized raw honey samples were liquefied at 40 °C in a water bath (manufacturer: Memmert GMBH – Schwabach, Germany). Homogenized samples were filtered through a double-layered gauze to retain the biggest impurities prior to the analysis.

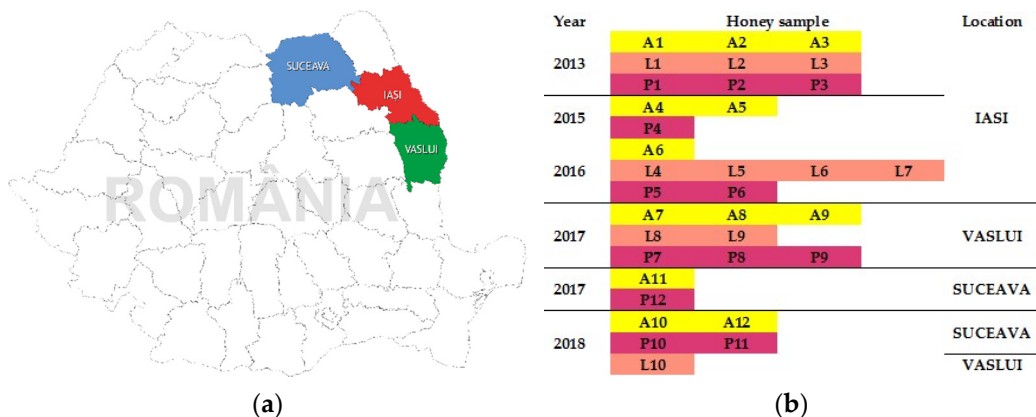

(a) (b)

**Figure 1.** (**a**) Locations of honey production; (**b**) year, type and location of studied honey samples.



### 2.1. Water-Insoluble Solids

The gravimetric method was applied to assess the water-insoluble solids (WIS). Ten grams of the homogenized honey sample was weighed on the PI-214 DENVER analytical scales (manufacturer: Denver Instrument GMBH – Gottingen, Germany). Prior to filtering through a qualitative-type filter paper adjusted prior to constant mass, the sample was diluted with distilled water. The sample was dissolved in distilled water and filtered through qualitative filter paper adjusted prior to constant mass. After several washes, the filter paper with the water-insoluble solids was dried in an ESAC 100 oven (manufacturer: S.C. Electronic April Aparatură Electronică Specială S.R.L. – Cluj-Napoca, Romania) at 105 °C till reaching constant mass, and then it was weighed on the analytical balance. The content of water-insoluble solids was calculated by the difference between the filter paper with water-insoluble solids weight and filter paper weight and was expressed as a percentage [10,12]

### 2.2. Color

Honey pigmentation was spectrophotometrically analyzed at 635 nm wavelength, using the Shimadzu UV-1700 Pharma Spec instrument (manufacturer: Shimadzu Corporation, Analytical Instruments Division, Kyoto, Japan) from 50% (*w/v*) honey aqueous solutions after honey samples were centrifuged at 3200 rpm for 5 min [13–15] in the UNIVERSAL 320 centrifuge (manufacturer: Hettich GMBH –Tuttlingen, Germany). The absorbance units were converted in mm Pfund using the relation

$$\text{Pfund (mm)} = -38.7 + 371.39 \times \text{Abs} \tag{1}$$

in which Pfund = honey color on the Pfund scale (mm); Abs = the value of the absorbance read at the wavelength of 635 nm.

### 2.3. Refractive Index, Moisture and Solid Substances

Raw honey refractive index measurements were carried on the ABBÉ AR 2008 refractometer (manufacturer: Kruss Scientific GMBH, Hamburg, Germany) (distilled water procedure was used to calibrate the device). The temperature of samples was determined using an EKT Hei-Con temperature sensor (manufacturer: Heidolph GMBH, Schwabach, Germany) and the refractive index, read on a refractometer, was corrected by adding 0.00023 for every 1 °C above 20 °C. Moisture content was then identified in the table of correspondence between water content and refractive index values at 20 °C [10,12,16]. Moisture content was expressed as a percentage. Solid substances, expressed as %, were calculated with the formula

$$\text{SS (\%)} = 100 - \text{M} \tag{2}$$

where SS = solid substances (%); M = moisture sample (%).

### 2.4. Total Soluble Solids

The total soluble solids were represented by total soluble sugars. The amount of total soluble solids, expressed as Brix degrees (a percentage of sugar is considered to be, at 20 °C, one °Brix) was read from the table of correspondence between the refractive index at 20 °C and degrees Brix [17].

### 2.5. Specific Gravity

The gravimetric method was applied to assess the raw honey specific gravity, using the pycnometer device. Values were obtained by dividing the pycnometer mass (bottle of 50 mL) filled with honey by the mass of the same bottle, filled with distilled water [18,19].

## *2.6. pH and Free Acidity*

Honey solution 10% (*w/v*) was measured on a MULTI 3320 multiparameter (manufacturer: WTW GMBH, Weilheim, Germany) to achieve pH values [12,19], while free acidity was assessed by using the titration method on the same solution, with 0.1 N NaOH (purity 99.6%, purchased from Lachner—Czech Republic), and the values were expressed in meq kg$^{-1}$ [20].

## *2.7. Ash*

Assessment of ash content in honey samples was carried out by the calcination method (550 °C) in the Nabertherm B180 furnace (manufacturer: Nabertherm GMBH, Lilienthal, Germany); the results were expressed in g/100 g [10,20,21].

## *2.8. Electrical Conductivity*

Electrical conductivity values were obtained by measuring 20% honey solution, calculated on dry matter, with the MULTI 3320 multiparameter (manufacturer: WTW GMBH, Weilheim, Germany). The solution was made with ultrapure water produced by the Barnstead EASY PURE II system (manufacturer: Thermo Fisher Scientific co. ltd., Iowa, USA); the results were expressed in μS cm$^{-1}$ [12,19,20].

## *2.9. Statistical Analyses*

All analyses were carried out in triplicate, firstly, to achieve the main descriptive statistics (mean ± st. dev., variation coefficient), followed by one-way ANOVA ($p < 0.05$). Pearson's correlation testing was run (r) to highlight causalities between studied parameters of honey samples. IBM SPSS Statistics14.0 version was used for principal component analysis (PCA) to visually observe Pearson's correlation [22]. For hierarchical cluster analysis (HCA), centroid linkage was used to discriminate the honey samples taking into account the values of the parameters determined in this study.

## 3. Results

### *3.1. Water-Insoluble Solids*

Water-insoluble solids content ranged within 0.023 and 0.131% in all mixed samples and varied from 0.023% to 0.122% in acacia honey, from 0.032% to 0.109% in linden honey and from 0.036% to 0.131% in multifloral honey (Table 1). The highest average of water-insoluble solids content was 0.080%, in multifloral honey samples. There are significant differences of this parameter between samples of the same type of honey and between the three types of honey (Figure 2). Statistically significant differences were observed between the means of water-insoluble solids content of acacia and multifloral honeys ($p = 0.05$).

**Table 1.** Descriptive statistics of studied honey sample parameters (*n* = 34).

| Parameters | Acacia | | | Linden | | | Multifloral | | |
| --- | --- | --- | --- | --- | --- | --- | --- | --- | --- |
| | **Descriptive Statistics** | | | | | | | | |
| | **Min–Max** | **Mean ± SD** | **CV%** | **Min–Max** | **Mean ± SD** | **CV%** | **Min–Max** | **Mean ± SD** | **CV%** |
| WIS (%) | 0.023–0.122 | 0.062 ± 0.03 | 48.86 | 0.032–0.109 | 0.070 ± 0.03 | 35.84 | 0.036–0.131 | 0.080 ± 0.03 | 41.39 |
| Pfund value (mm) | 0.3–8.8 | 2.2 ± 2.66 | 123.8 | 22.6–76.4 | 40.2 ± 15.17 | 37.77 | 15.4–70.1 | 44.8 ± 17.45 | 38.95 |
| Color intensity | water white-extra white | | | white-light amber | | | extra white-light amber | | |
| RI | 1.485–1.496 | 1.491 ± 0.00 | 0.26 | 1.488–1.499 | 1.493 ± 0.00 | 0.2 | 1.485–1.497 | 1.492 ± 0.00 | 0.25 |
| M (%) | 16.20–20.52 | 18.11 ± 1.47 | 8.13 | 15.20–19.32 | 17.42 ± 1.14 | 6.52 | 15.83–20.77 | 17.93 ± 1.44 | 8.03 |
| SS (%) | 79.48–83.80 | 81.89 ± 1.47 | 1.8 | 80.68–84.80 | 82.58 ± 1.14 | 1.38 | 79.23–84.17 | 82.07 ± 1.44 | 1.75 |
| TSS (°Brix) | 78.08–82.27 | 80.40 ± 1.43 | 1.78 | 79.20–83.26 | 81.07 ± 1.11 | 1.37 | 77.83–82.67 | 80.60 ± 1.41 | 1.75 |
| SG (g/cm$^3$) | 1.414–1.443 | 1.430 ± 0.01 | 0.68 | 1.422–1.450 | 1.435 ± 0.01 | 0.54 | 1.414–1.446 | 1.431 ± 0.01 | 0.66 |
| pH | 3.673–5.503 | 4.272 ± 0.45 | 10.57 | 3.742–5.398 | 4.479 ± 0.42 | 9.48 | 3.761–4.468 | 4.117 ± 0.21 | 5.13 |
| FA (meq kg$^{-1}$) | 2.4–17.0 | 8.6 ± 5.14 | 59.74 | 12.7–49.8 | 24.7 ± 11.88 | 48.05 | 15.6–50.0 | 27.1 ± 11.34 | 41.85 |
| Ash (%) | 0.030–0.242 | 0.088 ± 0.07 | 74.94 | 0.080–0.543 | 0.225 ± 0.13 | 59.43 | 0.088–0.277 | 0.178 ± 0.06 | 32.69 |
| EC (μS cm$^{-1}$) | 130–500 | 223 ± 12.31 | 50.4 | 279–646 | 506 ± 120.97 | 23.92 | 210–679 | 414 ± 162.27 | 39.16 |

WIS—water-insoluble matter. RI—refractive index. M—moisture. SS—solid substances. TSS—total soluble substances. SG—specific gravity. FA—free acidity. EC—electrical conductivity.

### *3.2. Color*

Honey pigmentation was variable, ranging from white to a light tint of amber. The lowest Pfund value was 0.3 mm Pfund in acacia honey (A7 sample) and the highest Pfund value was 76.4 mm Pfund in linden honey, sample L10. In Figure 3, it is observed that the linden and multifloral honey samples have different color shades, for $p < 0.05$. Statistically, there was a significant difference in mm Pfund means between the two types of honey: acacia and linden, and acacia and multifloral.

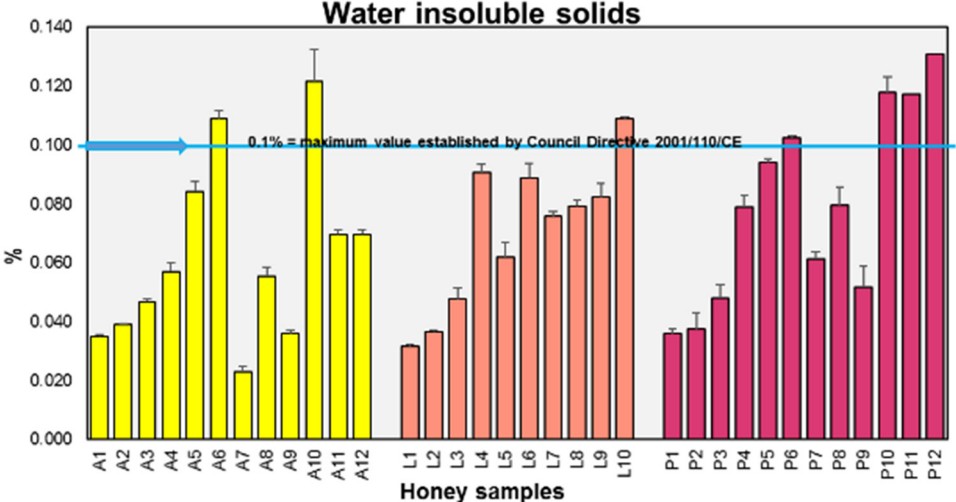

**Figure 2.** Average values of water-insoluble solids content of honey samples.

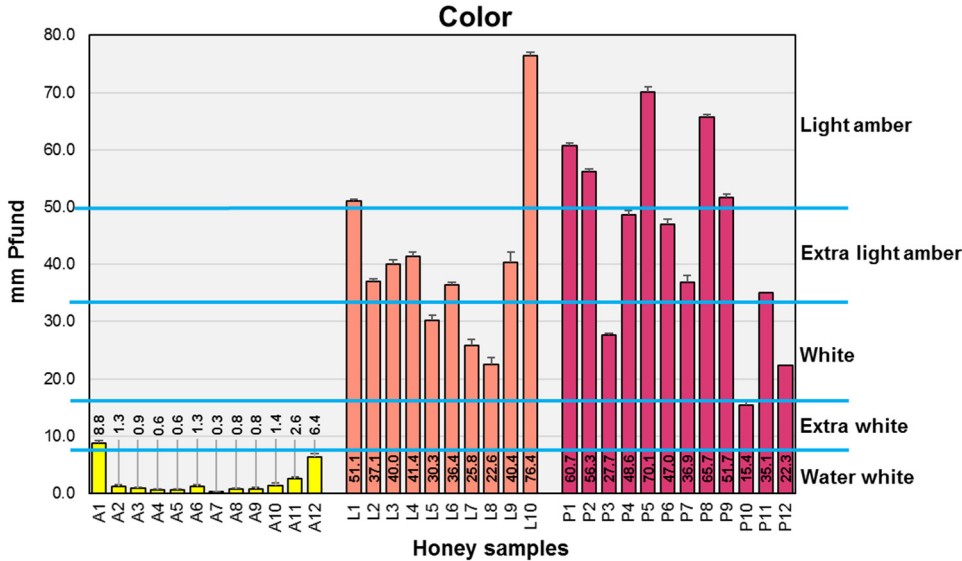

**Figure 3.** Average values of color intensity of honey samples.

### *3.3. Refractive Index and Moisture*

The minimum refractive index value of 1.485 was registered in one sample of acacia honey (A6 sample) and in one multifloral honey (P10 sample), while the maximum value of 1.499 was registered in a linden honey sample (L7 sample) (Table 1). There were no significant differences in average refractive index values between the analyzed honey types.

The moisture of investigated samples varied between 15.20 and 20.77% (Table 1). The highest average moisture values were measured in acacia and multifloral samples, i.e., 20.40% in A11, 20.52% in A12 and 20.77% in P10 (Figure 4). Acacia and linden honey groups differed significantly ($p < 0.05$), while the significance threshold was not exceeded for the paired comparisons of acacia and multifloral groups, and linden and multifloral groups.

### 3.4. Solid Substances and Total Soluble Substances

The solid substances values ranged between 79.23% and 84.80%. The highest and lowest average values were in linden honey samples, and values of 82.58% and of 81.89% were found in acacia honey samples (Table 1). Acacia and linden groups differed significantly ($p < 0.05$) in terms of solid substances.

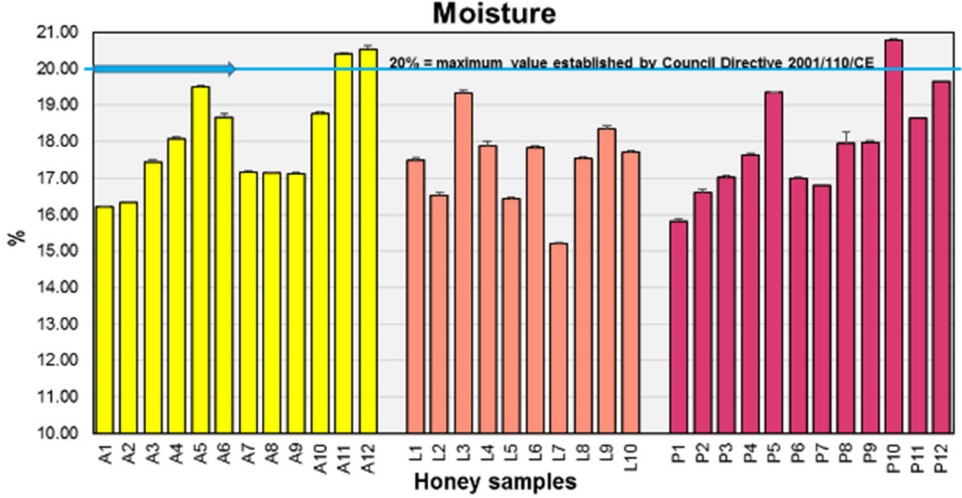

**Figure 4.** Average values of moisture content of honey samples.

Linden samples revealed the highest content of total soluble substances while the lowest one measured was found in the acacia honey group (80.40 °Brix). The maximum value of total soluble substances was 83.26 °Brix in one linden sample, L7, and the minimum value of total soluble substances was 77.83 °Brix in one linden sample, P10. A statistically significant difference ($p < 0.05$) for total soluble substances was found between average values of acacia and linden groups.

### 3.5. Specific Gravity

The average values of specific gravity were measured between 1.414 and 1.450 g/cm$^3$ (Table 1) and acacia and linden groups differed significantly ($p < 0.05$).

### 3.6. pH and Free Acidity

pH values ranged between 3.675 in acacia honey samples and 5.398 in linden honey samples (Table 1).

The lowest free acidity was measured in acacia samples (8.6 meq kg$^{-1}$), while the highest one, 27.1 meq kg$^{-1}$, was measured in multifloral honey samples. Figure 5 shows that there are differences between free acidity in honey samples in the same type of honey and between different types of honey. A statistically significant difference ($p < 0.05$) was found between acacia versus linden and multifloral samples.

### 3.7. Ash and Electrical Conductivity

An average value of 0.088% ash was obtained in acacia honey samples and of 0.225% in linden honey samples (Table 1). Electrical conductivity had the lowest average of 223 μS cm$^{-1}$ in acacia samples, then increased in multifloral ones to 414 μS cm$^{-1}$, to reach, eventually, 506 μS cm$^{-1}$ in linden samples. Figure 6 shows that there are differences ($p < 0.05$) between ash and electrical conductivity in honey samples in the same type of honey and between different types of honey.

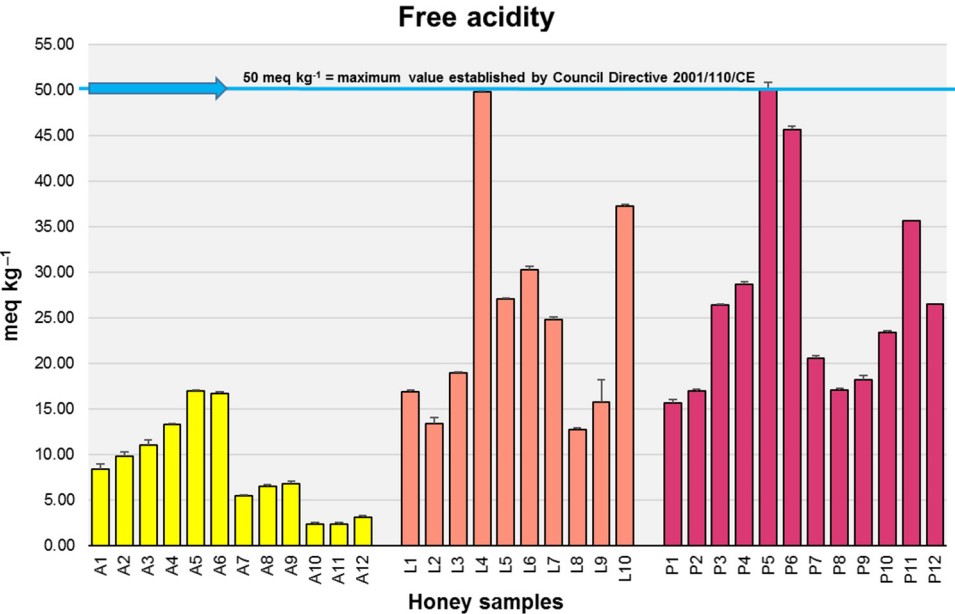

**Figure 5.** Free acidity values in acacia, linden and multifloral samples.

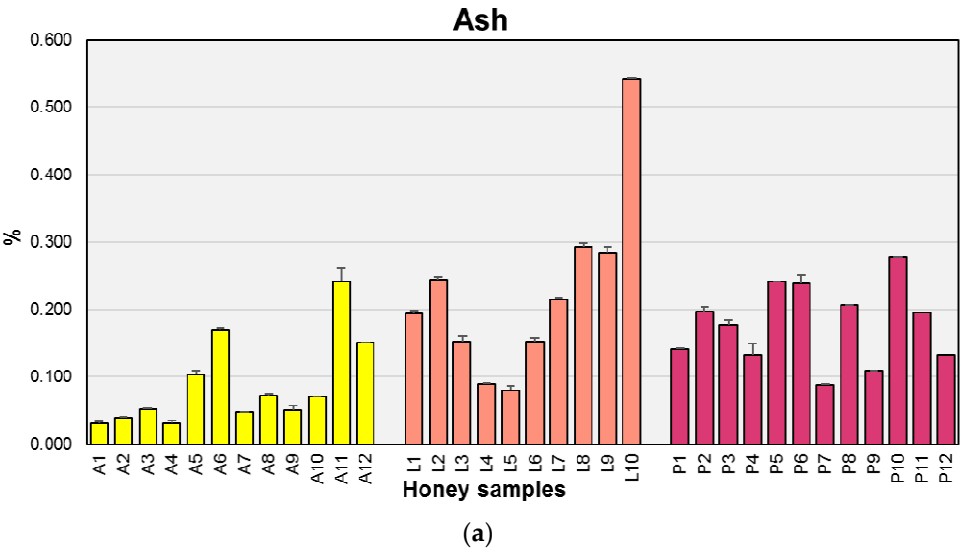

(**a**)

**Figure 6.** *Cont.*

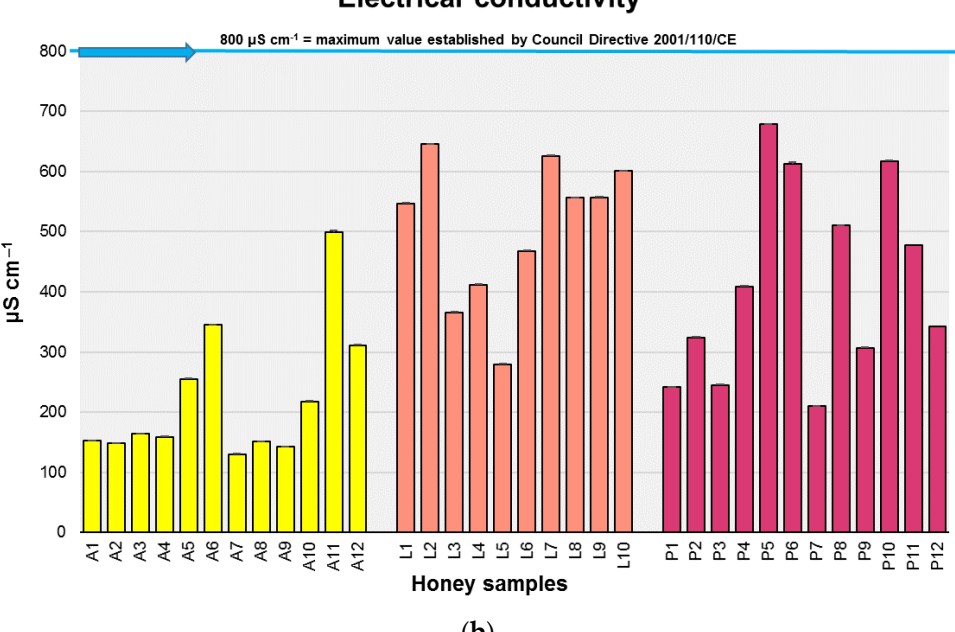

**(b)**

**Figure 6.** (**a**) Average values of ash content of honey samples; (**b**) average values of electrical conductivity of honey samples.

*3.8. Statistical Analyses*

Figure 7 depicts the identified correlations between the main physical and chemical investigated parameters.

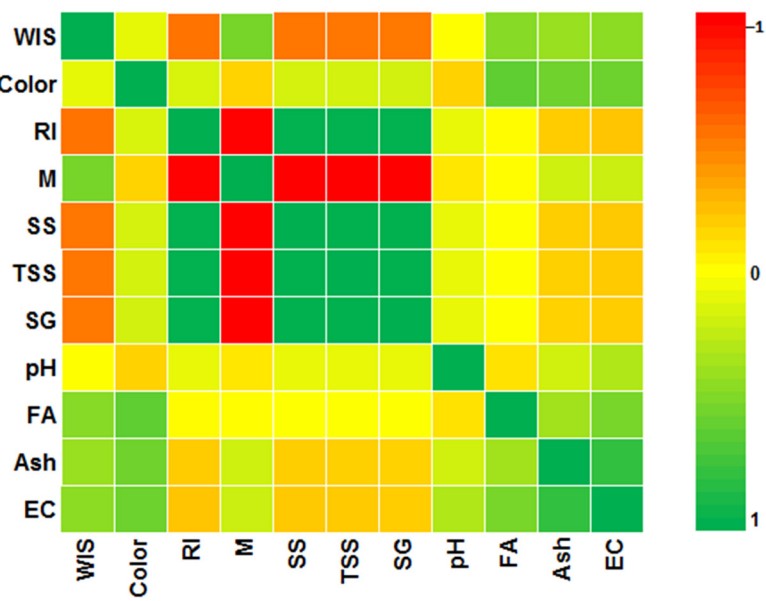

**Figure 7.** Head map of Pearson's correlation between investigated honey parameters.

A positive strong correlation between ash and electrical conductivity (r = 0.81) was found; refractive index was strongly positively correlated with three parameters: specific gravity (r = 0.99), total soluble substances (r = 1), solid substances (r = 1); solid substances had a strong positive correlation with two parameters: total soluble substances (r = 1) and specific gravity (r = 1), while total soluble substances had a strong positive correlation with specific gravity (r = 1).

In Figure 8, correlations between the studied parameters can be viewed. Two groups of correlated parameters can be observed: the first group with strong correlations contains refractive index, solid substances, total soluble substances and specific gravity, and the second group contains four parameters: color (mm Pfund), electrical conductivity, ash and free acidity.

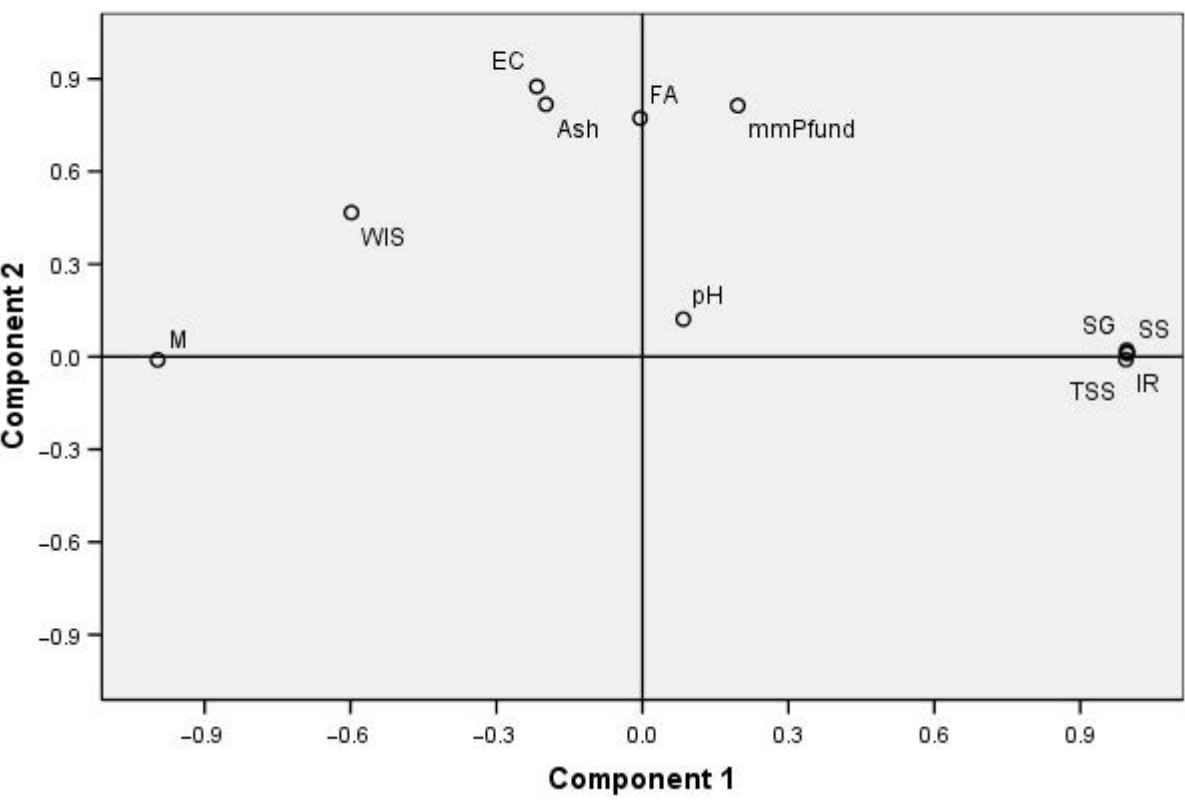

**Figure 8.** Principal component analysis (PCA) score plot for the analyzed honey samples.

The dendrogram (Figure 9) reveals the formation of four clusters in which we found honey samples with similar characteristics.

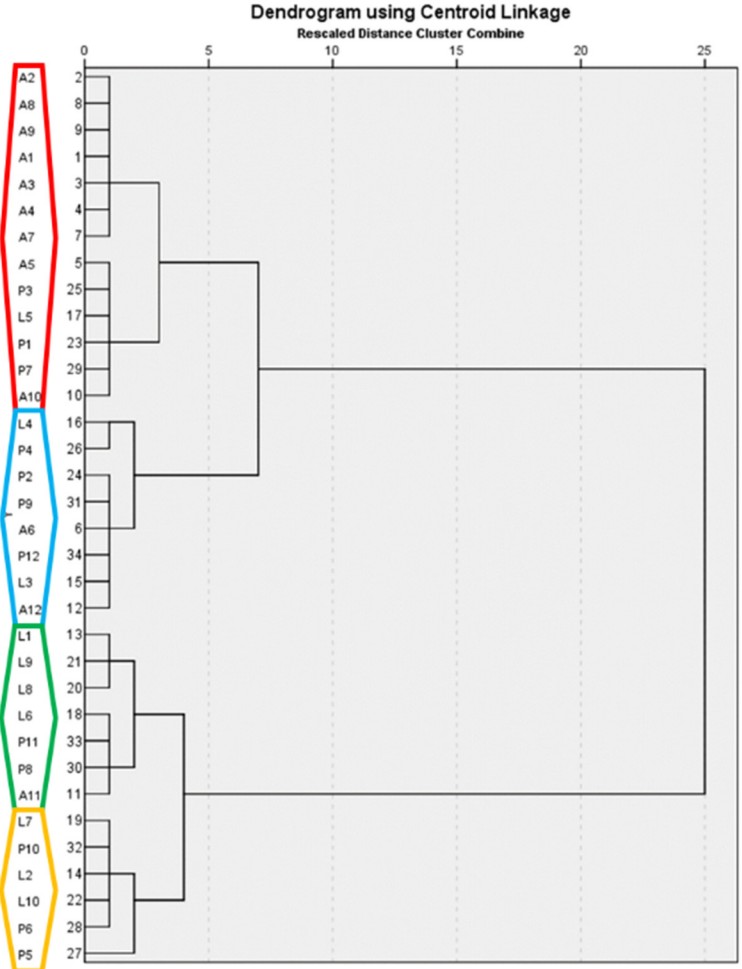

**Figure 9.** Hierarchical cluster analysis (HCA) of samples based on honey parameters.

## 4. Discussion

Each analyzed parameter is important and contributes to the appreciation of honey quality.

To produce commercially available honey, beekeepers carry out some operations such as extraction by pressing and centrifugation, filtration or storage [22]. The water-insoluble solids remaining in honey after those processes come from organic or inorganic sources such as wax fragments, small pieces of bees' bodies, bee larvae, plant particles such as wood, propolis, pollen, earth and dust [20,23]. Visually, for consumers, the clarity of honey is a quality criterion, so the amount of water-insoluble solids which give clarity to honey must be lower.

In accordance with the legislation for honey quality, the maximum acceptable content of water-insoluble solids in commercial honey is 0.1% and is tolerated up to 0.5% for raw honey [10,11]. Water-insoluble solids content in 20.59% of the total analyzed samples exceeded the 0.1% legal threshold (Figure 2). In 47.06% of the total honey samples, it reached values between 0.05% and 0.1%, while 32.35% of honey samples had less than 0.005% water-insoluble solids content. Such values suggest that some beekeepers run more careful filtration operations.

Studies made on Libyan honey, honey from the Czech Republic market and on honey collected from the western part of Turkey showed a content of water-insoluble solids under 0.1% [22,24–26]. High values of this parameter were found in honey samples from Brazil of 0.27–0.95% [27].

The filtering operation run to achieve a clear honey can lead to the loss of a quantity of pollen, a natural constituent of honey, with nutritional characteristics that increase the nutritional and therapeutic properties of honey [28,29]. Considering that the honey samples studied are not for sale as processed commercial products, this parameter indicates a good product quality.

The qualitative parameters of honey, as it is known, are influenced by many factors; therefore, this product is intensively studied in each country. Various published studies (Table 2) have shown differences in quality between samples of honey from different countries and even from different geographical areas of the same country.

**Table 2.** Physico-chemical properties of linden, acacia and multifloral honey in the present study and in several previous studies.

| References | Country | M (%) | TSS (Brix°) | SG (g cm$^{-1}$) | pH | FA (meq kg$^{-1}$) | EC (mS cm$^{-1}$) | Ash (%) |
|---|---|---|---|---|---|---|---|---|
| | | | | Linden | | | | |
| Present study | Romania | 15.20–19.32 | 79.20–83.26 | 1.422–1.450 | 3.742–5.398 | 12.70–49.80 | 0.279–0.646 | 0.080–0.543 |
| Mărghitaș et al. (2009) | Romania | 16.70–19.10 | - | - | - | - | - | 0.190–0.300 |
| Stihi et al. (2016) | Romania | 17.20–18.80 | - | - | 3.840–4.350 | - | 0.202–0.346 | - |
| Popescu et al. (2015) | Romania | 5.40–6.00 | 78.70–82.30 | - | 3.600–4.700 | - | 0.410–0.730 | - |
| Purcarea et al. (2016) | Romania | 18.44 | - | - | 4.425 | - | 0.486 | 0.186 |
| Scripcă et al. (2019) | Romania | 16.00–17.90 | - | - | - | - | 0.550–0.720 | - |
| Kędzierska-Matysek et al. (2018) | Poland | 19.40 | 79.20 | - | 4.130 | 14.50 | 0.579 | - |
| Tomczyk et al. (2019) | Poland | - | - | - | 3.810 | 34.20 | 0.530 | - |
| Lazarević et al. (2012) | Serbia | 13.41–22.48 | - | - | 3.980–5.400 | 8.20–26.20 | 0.300–0.760 | - |
| Matović et al. (2018) | Serbia | 17.20 | | | 3.980–5.400 | 12.18 | 0.390 | 0.060 |
| Tomczyk et al. (2019) | Slovakia | 18.35 | - | - | 3.900 | 21.60 | 0.230 | - |
| | | | | Acacia | | | | |
| Present study | Romania | 16.20–20.52 | 78.08–82.27 | 1.414–1.443 | 3.673–5.503 | 2.40–17.00 | 0.130–0.500 | 0.030–0.242 |
| Stihi et al. (2016) | Romania | 16.70–22.80 | - | - | 3.650–4.630 | - | 0.097–0.268 | - |
| Mărghitaș et al. (2009) | Romania | 16.60–19.80 | - | - | - | - | - | 0.030–0.280 |
| Mărghitaș et al. (2010) | Romania | 17.20–19.00 | - | - | 3.860–4.090 | 1.84–10.87 | 0.098–0.212 | - |
| Cimpoiu et al. (2013) | Romania | - | - | - | - | - | - | 0.040–0.270 |
| Popescu et al. (2015) | Romania | 3.90–6.20 | 78.30–83.60 | - | 3.700–4.300 | - | 0.110–0.270 | - |
| Scripcă et al. (2019) | Romania | 16.30–18.00 | - | - | - | - | 0.220–0.350 | - |
| Ahamed et al. (2017) | Saudi Arabia | 8.80–13.85 | - | 1.420–1.460 | 3.320–3.770 | 19.80–65.00 | 0.364–1.207 | 0.020–0.200 |
| Attri (2011) | India | 17.07–17.20 | - | 1.426–1.440 | - | - | 0.240–0.380 | 0.010–0.100 |
| Lazarević et al. (2012) | Serbia | 13.90–20.57 | - | - | 3.490–5.850 | 7.80–29.60 | 0.100–0.680 | - |
| Tomczyk et al. (2019) | Poland | 17.73 | - | - | 3.790 | 25.60 | 0.420 | - |
| Tomczyk et al. (2019) | Slovakia | 17.86 | - | - | 3.710 | 16.10 | 0.200 | - |
| | | | | Multifloral | | | | |
| Present study | Romania | 15.83–20.77 | 77.83–82.67 | 1.414–1.446 | 3.761–4.468 | 15.60–50.00 | 0.210–0.679 | 0.088–0.277 |
| Pauliuc et al. 2020 | Romania | 19.60 | - | - | 4.090 | 23.90 | 0.354 | - |
| Popescu et al. 2015 | Romania | 4.80–7.40 | 80.20–83.00 | - | 3.200–4.600 | - | 0.232–0.831 | - |
| Cimpoiu et al. 2013 | Romania | | - | - | - | - | - | 0.050–0.100 |
| Scripcă et al. 2019 | Romania | 15.90–17.90 | - | - | - | - | 0.280–0.450 | - |
| Aazza et al. (2013) | Portugal | 19.00–19.90 | - | - | 3.750–3.870 | 28.62–39.75 | 0.358–0.469 | 0.170–0.330 |
| Anguebes et al. (2016) | Mexico | 14.15–18.94 | 82.44–86.06 | - | 3.800–4.400 | 15.77–23.03 | 0.580–0.680 | 0.120–0.160 |
| Halouzka et al. (2016) | Czech Republic | 17.00–18.20 | - | - | 3.870–4.390 | 15.70–17.80 | 0.270–0.670 | - |
| Cantarelli et al. (2008) | Argentine | 14.28–18.60 | - | - | 3.340–4.700 | 21.23–43.20 | - | 0.060–0.210 |
| Giorgi et al. (2011) | Italy | 16.30–17.30 | - | - | 3.850–4.510 | 9.00–29.00 | 0.150–0.780 | - |
| Kędzierska-Matysek et al. (2018) | Poland | 18.40 | 80.10 | - | 4.730 | 4.50 | 0.371 | - |
| Tomczyk et al. (2019) | Poland | 18.65 | - | - | 3.570 | 37.00 | 0.350 | - |
| Khalil et al. (2012) | Algeria | 11.59–14.13 | - | - | 3.700–4.000 | - | 0.417–0.806 | - |
| Krishnasree and Ukkuru (2017) | India | 13.30–14.38 | 76.00–77.50 | 1.380–1.400 | 3.730–3.830 | 31.00–32.00 | 0.130–0.310 | 0.140–0.160 |
| Almeida et al. (2016) | Brazil | 17.46–20.28 | - | - | 3.650–4.170 | 12.77–48.04 | - | 0.075–0.472 |
| Tomczyk et al. (2019) | Slovakia | 18.53 | - | - | 3.680 | 23.30 | 0.210 | - |
| Kahraman et al. (2010) | Turkey | 15.30–16.90 | - | - | - | 23.90–24.40 | - | 0.280–0.290 |

M—moisture. SS—solid substances. TSS—total soluble substances. SG—specific gravity. FA—free acidity. EC—electrical conductivity.

The first thing the consumer does with honey is the visual inspection and the color is an attractive attribute, having a strong role in the price setting, the acceptability and the degree of purchase of the product [30]. Acacia samples' pigmentation ranged from waterish white to extra white, and the linden honey color oscillated between white and light amber, while in multifloral samples, the color had darker shades from extra white to light amber. The darkest colors, light amber, were found in 20.59% out of all samples (seven

samples of honey); eleven samples of acacia honey (29.41% of the total studied samples) had the lightest color, water white color.

Many studies investigated the chromatic of acacia, linden and multifloral honey. Some studies reported different mm Pfund values between 11 and 219.42 mm Pfund [15,31–35]. Pigmentation was reported as positively correlated with certain organic compounds in honey (total phenolic and flavonoids), as well as with their induced subsequent antioxidant activity [14,19,36–38]. Further, dark-tinted honey seems to be richer in minerals than light honey [1,32,39].

The type of nectar as well as its collection period, water, air, soil of each geographical region, processing operations and lengths and conditions of storage are responsible for the variety of colors [40,41]. In our findings, the color of raw honey samples (linden and multifloral) indicated good quality in terms of nutrition.

The refractive method is the most used method because it is easy to carry out and it can be reproduced without difficulty. Refractive index is a parameter whose value is correlated to and dependent on the values of other parameters, such as water content, solids substances, total soluble solids and specific gravity. The higher the refractive index, the lower the moisture content. Low values of moisture content prevent fermentation of honey and increase shelf life and also the storage time [42]. Less than 20% water content is recommended to be in honey, according to international regulations, transposed in Romanian standards [10,11]. In this study, three honey samples (8.82% of total) with a higher moisture content than the maximum legally allowed limit of 20% were identified. However, if kept in optimal storage conditions, in a room with low humidity, at 10 °C, away from sunlight, the conservation status is improved and there is no risk of fermentative process occurrence. The highest range of water content was found in multifloral honey, from 15.83% to 20.77%. Studies have shown that the moisture content had a large interval of values, from 4.8% to 22.8%, in multifloral Romanian honey [32,43–45]. Moisture content ranged from 14.15% to 18.94% of honey samples from Mexico and Argentina [21,46]. The values of moisture content in acacia honey from Serbia found by [47] fell within the 13.90–20.57% interval and between the 13.41 and 22.48% limits in linden honey samples. To prevent mold growth in honey, in order to improve preservation and storage, it is compulsory to assess and monitor the moisture content [48].

Solids reversely correlated with moisture. The higher the solids level, the richer the content of sugars and minerals. Linden honey samples had the highest average solid substances, 82.58%, and the poorest level of moisture, 17.42%.

Most of the total soluble solids in honey are sugars. In general, the amount of total soluble solids is 80% or above [49]. According to the grading system of the United States Department of Agriculture, when results exceed 80 °Brix (<20% water), honey is qualitative and has better stability during storage, as moisture decreases when the total soluble solids level increases [17]. This classification refers, in particular, to an additional insurance in terms of water content (total soluble solids are reversely correlated with water and with safety during storage for a longer period). Although, in this research, ten honey samples had a total soluble solids content between 77.83% and 80%, less than the established value of 80% by the USDA, their nutritional quality is high and relatively stable during storage.

The specific gravity of honey is inversely related to its moisture; the denser a honey, the less humidity it will possess [50]. The lowest average specific gravity was measured in acacia honey (1.430 g/cm$^3$), while the highest one (of 1.435 g/cm$^3$) was measured in linden samples. Quite close values, between 1.426 and 1.440 g/cm$^3$, were found by Attri [50] in Indian honey and between 1.42 and 1.46 g/cm$^3$ by Ahamed et al. [18] in honey from Saudi Arabia. Lower values were found, 1.40 g/cm, by [49] in Indian raw honey. It is important to know the specific gravity because this parameter can reveal whether or not the honey is counterfeit and whether it has a practical significance in keeping track of the amount of honey stored [20,23].

While most of the honey varieties have pH values between 3.5 and 5.5, this parameter is not legislated. The pH values we measured fell within the above interval. Lower pH

values guarantee bacteriostatic and fungi static effects, rendering this alimentary raw matter more fit to be included in processed food products, due to the higher acidity which creates asepsis [51]. Further, acidity is relevant during honey extraction and preservation because it will affect the subsequent sensorial properties via texture, overall stability and minimal durability as a marketed product [37,49,52]. Specifically, the investigated samples had pH values that oscillated between 3.673 and 5.398. Other pH values, from 3.65 to 4.70, have been reported for Romanian honey [43,45,53,54].

Honey is an acid food. Acidity is influenced by chemical properties of the organic and inorganic acids. Further, the action of glucose oxidase could induce alterations of honey acidity via gluconic acid accumulation [19]. Moreover, other biochemical species, such as lactones, esters and even inorganic acids, could influence the concentration of lactic acid in honey, influencing thus its free acidity level [32]. This parameter is important because its value is a measure of honey freshness. All of the free acidity values from the investigated honey samples were lower than 50 milliequivalents acid per 1000g, the maximum allowed value specified by the legislation (European Commission, 2002). The lowest free acidity was obtained in an acacia sample (A10 sample). Figure 5 reveals the lowest values of acacia samples, between 2.4 and 17.0 meq $kg^{-1}$. The highest value of 27.1 meq $kg^{-1}$ was measured in multifloral honey, and then intermediate acidity was observed in linden samples (24.7 meq $kg^{-1}$). In other studies, free acidity values were close to our findings: 21.6 meq $kg^{-1}$ in Slovakian linden honey [55], 7.8–29.6 meq $kg^{-1}$ in Serbian acacia honey [56] and 28.62–39.75 meq $kg^{-1}$ in multifloral Portuguese honey [56].

Ash and its positively correlated conductivity are traits used to differentiate blossom honey from honeydew honey [2]. Ash content is given by the macro- and micro-elements of honey, which fall under the influence of the composition of collected nectar. Minerals are important nutritional compounds, with relevance in consumers' health and welfare [1]. Blossom honey is poorer as a mineral content, which ranged between 0.02 and 0.3%. Variation limits of 0.03% and 0.277% were found in our original investigations. Linden honey had the richest ash content, 0.225%, with individual samples ranging from 0.080% to 0.543%; acacia honey had the lowest average ash content, of 0.088%. Other researchers reported ash content in Romanian honey from 0.03% to 0.08% in acacia, from 0.05% to 0.10% in multifloral honey [31,54] and 0.186% in the linden sort [44]. In Argentine multifloral honey, ash content was found between 0.06% and 0.21% [21], while Kahraman et al. [57] found 0.28–0.29% in Turkey honey. The amount of ash in investigated samples confirms the qualities of honey, knowing that they contain many essential mineral elements that play an important role in human metabolism.

Electrical conductivity (EC) indicates the property of a generic material to conduct an electric current. In honey, it is straightly influenced by the nectar botanical origin, as well as by other inorganic or organic molecules that could act as electrolytes, such as ions, organic acids and proteins [40]. Directive 2001/110/EC of the European Union (amended in 2007) specifies the upper level of allowed electrical conductivity for honey of nectar, 0.8 µS $cm^{-1}$, with an exception in chestnut honey which should be above 0,8 µS $cm^{-1}$ [11]. The found electrical conductivity ranged between 130 in acacia honey and 679 µS $cm^{-1}$ in multifloral honey. The highest average value of 506 µS $cm^{-1}$ was found in linden honey.

A variability of values of this parameter was observed within the same type of honey: in acacia, values ranged from 130 to 500 µS $cm^{-1}$, and in linden between 279 and 646 µS $cm^{-1}$, while in multifloral honey, the values were measured between 210 and 679 µS $cm^{-1}$. Pretty similar dynamics of values were reported in Romanian samples: between 202 and 730 µS $cm^{-1}$ in linden honey [5,43–45,58]; between 97 and 350 µS $cm^{-1}$ in acacia [5,43,45,49,58]; and between 232 and 831 µS $cm^{-1}$ in multifloral honey [32,39,45].

Table 2 reveals the results of some parameters, from other studies, i.e., honey samples from different countries [7,47,59–62].

All values of electrical conductivity measured in our study revealed that samples originated from blossom honey and the value of this parameter, closely related to organic acids and to mineral salts, is an indicator which contributes to quality.

A strong negative correlation between moisture and three parameters was calculated: solid substances, total soluble substances and specific gravity (r = −1). A moderate negative correlation was observed between water-insoluble substances and four other parameters: refractive index (r = −0.54), solid substances (r = −0.54), total soluble substances and specific gravity (r = −0.53). Mild positive correlations were calculated between moisture and water-insoluble substances (r = 0.54); between free acidity and electrical conductivity (r = −0.53); and between color (mm Pfund) and electrical conductivity (r = 0.57), ash (r = 0.56) and free acidity (r = 0.63). Strong positive correlations were identified between ash and electrical conductivity (r = 0.81), Figure 7; refractive index was strongly positively correlated with three parameters: specific gravity (r = 0.99), total soluble substances (r = 1), solid substances (r = 1); solid substances had a strong positive correlation with two parameters: total soluble substances (r = 1) and specific gravity (r = 1), while total soluble substances had a strong positive correlation with specific gravity (r = 1).

Our original findings are comparable with the results obtained in other investigations by Attri, Gomes et al. and El Sohaimy et al. [50,51,63], which found positive strong correlations between mineral content and conductivity (r = 0.93) and acidity (r = 0.75).

Krishnasree and Ukkuru [49] observed direct correlations between the pairs pH–moisture, pH–acidity and acidity–ash, and Majewska et al. [64] found a positive strong correlation between conductivity and ash, while Baloš et al. [9] found that conductivity is closely inter-related to the concentration of minerals and organic acids.

A PCA score plot was constructed for visualizing the relationship between physico-chemical parameters, and positive correlations between refractive index, solid substances, total soluble substances and specific gravity could be noticed. A correlation was also observed between four parameters: color (mm Pfund), electrical conductivity, ash and free acidity.

HCA showed similarities between honey samples based on the investigated honey parameters, and in Figure 9, four clusters are observed. There are some different types of honey samples with similarities to the studied parameters, as it is observed in the first cluster that comprises nine acacia, one linden and three multifloral honey samples. The second cluster has two acacia, two linden and four multifloral honey samples. The third cluster has one acacia, four linden and two multifloral honey samples. The last clusters have three linden and three multifloral honey samples.

## 5. Conclusions

Among all analyzed samples from eastern Romania, 79.42% were in accordance with the quality regulations for honey as a commercial product.

Those honey samples exceeding the moisture upper limit of 20% do not present a risk of fermentative alteration if they are conditioned in a specially prepared drying room to gradually remove water excess and then conditioned and prepared to be marketed.

Free acidity assessments underlined the acid characteristic of honey, especially in acacia sorts. The different values for the mineral content reveal the nectar was harvested from different floral regions in eastern Romania. The floral origin was also confirmed by electrical conductivity levels below 800 $\mu$S cm$^{-1}$.

Hierarchical cluster analysis revealed that honey samples belonging to different sorts could have close values of the physico-chemical traits.

The quality of honey is highly dependent on many factors such as the year of sample collection, variety of landforms and diversity of the flora, as the data of this study revealed, basing on the diversity of raw honey quality from east Romania.

**Author Contributions:** Conceptualization, A.A. and I.M.P.; methodology, A.A. and C.-G.R.-R.; software, A.A. and C.-G.R.-R.; validation, A.A. and I.M.P.; formal analysis, A.A., C.-G.R.-R., G.F. and G.N.; investigation, A.A., C.-G.R.-R., G.F. and G.N.; data curation, A.A. and I.M.P.; writing—original draft preparation, A.A. and C.-G.R.-R.; writing—review and editing, A.A., C.-G.R.-R. and I.M.P.; supervision, A.A. All authors have read and agreed to the published version of the manuscript.

**Funding:** This research received no external funding.

**Institutional Review Board Statement:** Not applicable.

**Informed Consent Statement:** Not applicable.

**Data Availability Statement:** The data presented in this study are available on request from the corresponding author.

**Conflicts of Interest:** The authors declare no conflict of interest.

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
