# Peer review of "Quality Assessment of Raw Honey Issued from Eastern Romania"

_agriculture, doi:10.3390/agriculture11030247_

Round 1
Reviewer 1 Report
The current work is very interesting as it documents the high quality of Romanian honeys which depends on several biotic and abiotic environmental factors. I have some suggestions for the authors to improve their paper:
a) line 54, check the repetition "are correlated"; line 377 I do no think that is possible to start the sentence like this "[50] noticed". Please, check the whole text for this mistakes.
b) the work is well-written and conducted; however, my major concern is that only 34 samples deriving from 3 regions have been investigated. This aspect strongly limits the works. To overcome this aspect, I strongly suggest the authors to add another section to the discussion where they compare the physico-chemical parameters obtained in their work with those reported in other similar papers. In particular, it would be interesting if the authors could make a comparison between the same typology of honeys produced in other world countries.. in order to better comment their data, to find other correlations, distinguish honey for geographic area, and even understand if and at what extent these parameters can be modulated. I report here several works that should be considered during this comparison: Environmental Science and Pollution Research, 2018, 25.14: 14148-14157; Food Chemistry, 2010, 123.1: 41-44; Journal of Food Composition and Analysis, 2003, 16.5: 613-619; International journal of food science & technology, 2010, 45.6: 1111-1118; Plant Biosystems 2017, 151.3: 450-463;Microchemical Journal, 2009, 93.1: 73-77; Journal of AOAC International, 2017, 100.4: 840-851; Foods, 2020, 9.3: 306.
c) Finally, I did not understand how did the authors identified the botanical origin of the samples (i.e., acacia, linden and polyfloral honeys)? Did someone make the melissopalynological analysis? How? Can the reader have the pollen spectra of these samples in the text, as supplemental material? The influence of minor blossoms and the percentage of purity of these honey can be fundamental for the interpretation of the chemico-physical parameters discussed in the text.
Author Response
Reviewer 1>
The current work is very interesting as it documents the high quality of Romanian honeys which depends on several biotic and abiotic environmental factors. I have some suggestions for the authors to improve their paper.
General answer: Thank you indeed for taking time and sharing your know-how in improving our paper and, as well, for your kind and professional suggestions. Hereby you'll find some punctual answers, while the paper was edited in detail, using track changes functions and comments were placed wherever necessary.
Reviewer 1 suggestions>
- a) line 54, check the repetition "are correlated"; line 377 I do no think that is possible to start the sentence like this "[50] noticed". Please, check the whole text for this mistakes. ANSWER> Checked and adjusted in all text. Thank you!
- b) the work is well-written and conducted; however, my major concern is that only 34 samples deriving from 3 regions have been investigated. This aspect strongly limits the works. To overcome this aspect, I strongly suggest the authors to add another section to the discussion where they compare the physico-chemical parameters obtained in their work with those reported in other similar papers. In particular, it would be interesting if the authors could make a comparison between the same typology of honeys produced in other world countries.. in order to better comment their data, to find other correlations, distinguish honey for geographic area, and even understand if and at what extent these parameters can be modulated. I report here several works that should be considered during this comparison: Environmental Science and Pollution Research, 2018, 25.14: 14148-14157 ; Food Chemistry, 2010, 123.1: 41-44 ; Journal of Food Composition and Analysis , 2003, 16.5: 613-619; International journal of food science & technology, 2010, 45.6: 1111-1118 ; Plant Biosystems 2017, 151.3: 450-463;Microchemical Journal, 2009, 93.1: 73-77 ; Journal of AOAC International, 2017, 100.4: 840-851; Foods, 2020, 9.3: 306. Answer: more comparative papers were cited and we built a new table for such comparisons, i.e. Table 2 in the Discussions session. Thank you for the suggestion!
- c) Finally, I did not understand how did the authors identified the botanical origin of the samples (i.e., acacia, linden and polyfloral honeys)? Did someone make the melissopalynological analysis? How? Can the reader have the pollen spectra of these samples in the text, as supplemental material? The influence of minor blossoms and the percentage of purity of these honey can be fundamental for the interpretation of the chemico-physical parameters discussed in the text.
Answer> The honey was collected straight from the beekeepers and it is raw honey. It was not thermally treated, so it was not a commercially marketable honey (as the ones we may find in big retailers). Beekeepers move their hives across large distances and camp in well known botanical massifs for nectar harvesting, during the flowering period of a major species. Thus, the bee families are dispatched nearby the massifs of acacia, or linden, or sunflower or on the meadows. The honey we sampled was produced from the nectar harvested during the maximum flowering phenophase of the specific seasonal melliferous culture (acacia or linden or sunflower and never overlapped). Towards the end of each harvesting seson, bees harvest from other species or from meadows and that honey is considered multifloral. We did not run pallinological analysis, knowing the actual source of nectar harvesting.
THANK YOU INDEED!

Reviewer 2 Report
The manuscript provides interesting data on a substantial number of Romanian honey samples, but there are some major issues that require authors' attention and various minor fauts that should be fixed.
Geographic reference to the East region of Romania correctly appears in the title, but in the text there the results are often extended to whole Romania. That should be avoided since the area sample came from is rather small if compared with whole Romania.
Authors cite several times the high quality of Romanian honeys, but their data do not support such a claim. There are too much samples with abnormal values.
Samples were collected in the 2013-2018 period and details on sample conservation are provided, but it would be interesting to know also when saples were analysed. Colour, pH and acidity change during conservation at room temperature even in the dark.
For the sake of clarity all analytical data should be presented in a single table. To keep such a table in a single page, colums and rows can be trasponed. Figures 2, 3, 4 and 5 are rather small and would greatly improve if enlarged; they can be pooled in two figures only, placing the graphs one over the other. Also figure 6b and figure 7 are a bit too littre.
The list of references is too long. Variuos reference are simply replicates of others and are not necessary to support the text. Reference to Crane's book could be enough in several instances.
The use of keywords different from the title would improve indexing the paper and could allow more citations.
Unifloral and multifloral honey should be used intread of monofloral and plyfloral honey.
Statistical significance is usually placed at P<0.05 and not P<0.5.
Herewith you find a list of minor hints:
line 19: 130-679 μS cm-1
line 27: wholesome instead of complete; honey is not a complete food since it mainly consistes of sugars
lines 33-34: diverse and very rich flora
line 34: there is a series of
line 37: linden instead of lime as it is written throughout the paper
line 39: polliniferous plants produce pollen, but honey bees collect nectar from nectariferous plants to produce honey
line 49: maximum 50 meq kg-1 free acidity,
line 52: nectar honey instead of honey of nectar
lines 54-55: are correlated twice
line 57: to which parameter are total soluble substances correlated?
line 68: highest purity reagents is rather generic; authors shoul be more precise
figure 1b: insert a line between 2015 and 2016; why sample A11 has been collected in 2017 and samples A10 and A12 in 2018?
line 76: was instead of were; paper is singular
line 95: sample moisture instead of moisture sample
line 98: 1% instead of a percentage
line 108: NaOH without )
line 111: Determination of ash content in honey samples was made ...
Line 114: measuring
lines 122-125: SPSS Statistic14.0 version (reference please!) was used for Principal camponent analysis (PCA) ...
Lines 146-150 deal with refractive index and shoul be placed under the 3.3. Refractive index and Moisture heading
line 154: A12 instead of A6; see figure 3
line 168: acacia honey group; see table 2
line 182: thre are differencies between ...
Lines 197-198: A positive strong correlation between ash and electrical conductivity (r = 0.81) has been found;
lines 198-202: refractive index is used to calculate moisture and obviously there is a perfect correlation between moisture and solid substances (see formula (2)!); also specifig gravity and total solubles substances are strictly correlated to moisture as is evidenced by PCA; you should not put so much emphasis on these results here and at lines 257-298 and 359-373
line 217: which is the difference between extraction and centrifugation?
lines 224-230 are not clearenough; they shoul be rewritten
lines 231-238: are references 24-27 the only references available? If not, why did you choose them?
lines 237-238: if the studied honey samples are not for market sale, what are they for?
lines 263-265: In this study,three honey samples ... legally allowed limit were identified.
line 265: define "optimal storage conditions"
line 268: a moisture content of 4.8% looks rather unusual; please check
line 272: IN in lower case
line 310: closed squre bracket and full point after 19
line 325: ... collected by bees. Minerals are important ...
line 354: honey
lines 386-392: you should explain why honeys from different sources cluster together
lines 934-936: conclusions are rather poor; you shoul try to improve them
Author Response
The manuscript provides interesting data on a substantial number of Romanian honey samples, but there are some major issues that require authors' attention and various minor faults that should be fixed.
Answer> Thank you indeed for dedicating your time and know-how in improving our paper. We will answer you punctually to your suggestions. Also, most of the punctual suggestions (i.e. corrections in the lines where you suggested) were operated in the manuscript. We used Track changes functions for reviewing and all modifications are accompanied by Comments on the right side of the manuscript pages. Hereby below we'll provide point by point answers to your questions>
Qa) Geographic reference to the East region of Romania correctly appears in the title, but in the text there the results are often extended to whole Romania. That should be avoided since the area sample came from is rather small if compared with whole Romania.
Authors cite several times the high quality of Romanian honeys, but their data do not support such a claim. There are too much samples with abnormal values.
Answer>Honey was sampled directly from the beekeepers, as raw honey. The quality traits were assessed on the raw honey and the results depict an appropriate level of quality for such type of honey, suggesting proper storage and lack of alteration. Beekeepers sell the raw honey to the big companies that run its further processing (heating at 38°C for liquefaction and water proportion decrease; removal of water-insoluble solids through filtering; quality traits investigation according to law) in order to become a commercially marketable product (en detailing for retail selling or for HoReCa-restoration-hospitality system). The quality traits of the raw honey we measured were the compared to the levels of quality traits expected for the commercially available honey. Therefore: 7 samples exceeded with more than 0.1% the insoluble solids level (maximal 0.131%); 3 samples presented moisture above 20% (maximal 20.77%); 1 single sample had the free acidity at the maximal allowed by the regulations, 50 meq kg-1. We consider the raw honey is tough qualitative, knowing it is taken by big companies for processing then selling within stable storage conditions and extended shelf life conditions.
Qb) Samples were collected in the 2013-2018 period and details on sample conservation are provided, but it would be interesting to know also when saples were analysed. Colour, pH and acidity change during conservation at room temperature even in the dark.
Answer>Raw honey, collected directly from the beekeepers was kept in glass jars, in in the dark at laboratory temperature (20±3°C). The analyses were run shortly after each collection and, of course, every sample was analysed the same year it was collected.
Q c) For the sake of clarity all analytical data should be presented in a single table. To keep such a table in a single page, colums and rows can be trasponed. Figures 2, 3, 4 and 5 are rather small and would greatly improve if enlarged; they can be pooled in two figures only, placing the graphs one over the other. Also figure 6b and figure 7 are a bit too littre.
Answer>Thank you for these precious suggestions. We reunited all results in a single table (Table 1) and enlarged all charts.
Q d) The list of references is too long. Variuos reference are simply replicates of others and are not necessary to support the text. Reference to Crane's book could be enough in several instances.
Answer>The first reviewer asked for supplemental references. We decided to keep the existing ones and included a few more, but we organized a comparison table between our findings and the literature ones, in order to have a more general overview (Table 2 in Discussions section).
Qe) The use of keywords different from the title would improve indexing the paper and could allow more citations.
We improved the keywords as you suggested, thank you!
Qf) Unifloral and multifloral honey should be used intread of monofloral and plyfloral honey.
Answer> proceeded with the modifications in all text, thanl you!
Qg) Statistical significance is usually placed at P<0.05 and not P<0.5.
Answer> of course, it is for 95%, P<0.05, we bag you apologies for such rookie <0.5 typing error, we adjusted for P<0.05 everywhere, thank you
Q+Herewith you find a list of minor hints:
Answer>we adjusted them punctually in the paper, track changes underline them. Also, we answered to the editor's request to rephrase some details in the text, due to plagiarism detection, so please lookup for Track changes and Comments in the right side of the documents which include Adjusted upon Reviewer 2 suggestion sentence. THANK YOU INDEED FOR THE PRECIOUS HELP!
line 19: 130-679 μS cm-1
line 27: wholesome instead of complete; honey is not a complete food since it mainly consistes of sugars
lines 33-34: diverse and very rich flora
line 34: there is a series of
line 37: linden instead of lime as it is written throughout the paper
line 39: polliniferous plants produce pollen, but honey bees collect nectar from nectariferous plants to produce honey
line 49: maximum 50 meq kg-1 free acidity,
line 52: nectar honey instead of honey of nectar
lines 54-55: are correlated twice
line 57: to which parameter are total soluble substances correlated?
line 68: highest purity reagents is rather generic; authors shoul be more precise
figure 1b: insert a line between 2015 and 2016; why sample A11 has been collected in 2017 and samples A10 and A12 in 2018?
line 76: was instead of were; paper is singular
line 95: sample moisture instead of moisture sample
line 98: 1% instead of a percentage
line 108: NaOH without )
line 111: Determination of ash content in honey samples was made ...
Line 114: measuring
lines 122-125: SPSS Statistic14.0 version (reference please!) was used for Principal camponent analysis (PCA) ...
Lines 146-150 deal with refractive index and shoul be placed under the 3.3. Refractive index and Moisture heading
line 154: A12 instead of A6; see figure 3
line 168: acacia honey group; see table 2
line 182: thre are differencies between ...
Lines 197-198: A positive strong correlation between ash and electrical conductivity (r = 0.81) has been found;
lines 198-202: refractive index is used to calculate moisture and obviously there is a perfect correlation between moisture and solid substances (see formula (2)!); also specifig gravity and total solubles substances are strictly correlated to moisture as is evidenced by PCA; you should not put so much emphasis on these results here and at lines 257-298 and 359-373
line 217: which is the difference between extraction and centrifugation?
lines 224-230 are not clearenough; they shoul be rewritten
lines 231-238: are references 24-27 the only references available? If not, why did you choose them?
lines 237-238: if the studied honey samples are not for market sale, what are they for?
lines 263-265: In this study,three honey samples ... legally allowed limit were identified.
line 265: define "optimal storage conditions"
line 268: a moisture content of 4.8% looks rather unusual; please check
line 272: IN in lower case
line 310: closed squre bracket and full point after 19
line 325: ... collected by bees. Minerals are important ...
line 354: honey
lines 386-392: you should explain why honeys from different sources cluster together
lines 934-936: conclusions are rather poor; you shoul try to improve them
